# Traffic Sign Recognition Evaluation for Senior Adults Using EEG Signals

**DOI:** 10.3390/s21134607

**Published:** 2021-07-05

**Authors:** Dong-Woo Koh, Jin-Kook Kwon, Sang-Goog Lee

**Affiliations:** 1Department of Media Engineering, Catholic University of Korea, 43 Jibong-ro, Bucheon-si 14662, Korea; metamo7@gmail.com; 2CookingMind Cop. 23 Seocho-daero 74-gil, Seocho-gu, Seoul 06621, Korea; k2gon.ug@gmail.com

**Keywords:** traffic sign recognition, electroencephalogram, brain computer interface, elderly drivers

## Abstract

Elderly people are not likely to recognize road signs due to low cognitive ability and presbyopia. In our study, three shapes of traffic symbols (circles, squares, and triangles) which are most commonly used in road driving were used to evaluate the elderly drivers’ recognition. When traffic signs are randomly shown in HUD (head-up display), subjects compare them with the symbol displayed outside of the vehicle. In this test, we conducted a Go/Nogo test and determined the differences in ERP (event-related potential) data between correct and incorrect answers of EEG signals. As a result, the wrong answer rate for the elderly was 1.5 times higher than for the youths. All generation groups had a delay of 20–30 ms of P300 with incorrect answers. In order to achieve clearer differentiation, ERP data were modeled with unsupervised machine learning and supervised deep learning. The young group’s correct/incorrect data were classified well using unsupervised machine learning with no pre-processing, but the elderly group’s data were not. On the other hand, the elderly group’s data were classified with a high accuracy of 75% using supervised deep learning with simple signal processing. Our results can be used as a basis for the implementation of a personalized safe driving system for the elderly.

## 1. Introduction

The reason elderly people have a high mortality rate in traffic accidents is due to their physical weakness. Deterioration in visual function due to aging mainly occurs in visual acuity, peripheral vision, visual acuity in dark places, contrast sensitivity, motion detection, and color discrimination [1]. The understanding of traffic signs of elderly drivers in a fast driving environment is highly related to safe driving, but the relevant studies are lacking. In particular, the elderly have a low driving frequency, but they are likely to lead to large-scale accidents. They are usually driving to meet basic needs such as market/hospital visits [2].

The elderly have various difficulties such as in judging the speed of other vehicles, changing lanes, etc. These problems are mainly caused by the elderly’s low cognitive ability, and the decline in their physical abilities such as impaired vision and neck rotation problems. For this reason, studies on the characteristics of the elderly and their reaction time in response to various driving events is very important for preventing traffic accidents [3]. Various studies are being conducted to overcome the less reactivity-related problem of elderly drivers, such as vehicle stop evaluation before a collision accident. Most of the other existing studies on elderly drivers are mainly evaluating whether or not they simply recognize the font size of the navigation and the traffic symbols [4]. Considering this point, it is necessary to evaluate the cognitive ability level of the elderly with various traffic signs that are often used while driving. Traffic signs include simple regulatory signs of the speed limit, indicate complex directions, symbols, and figures. Complex traffic signs can cause misunderstanding for elderly drivers, so intuitive traffic sign design becomes a very important factor in safe driving.

According to the recent development of vehicle convenience technology, the head-up display (HUD) device helps drivers to obtain traffic regulation and instruction information through various symbols and pictures without turning his or her head sideways while driving. However, due to the high visibility of HUD, excessive information that does not take into account the degree of understanding of the elderly driver can cause driving distraction. This indicates that the evaluation and recognition of the individual cognitive abilities of the elderly are important issues. The driving navigator should select various guidance methods in consideration of the cognitive state for the elderly driver to respond appropriately while driving. If the driver feels a lack of understanding of the driving navigation instructions, they should be guided by a stronger tone voice message or a longer warning window. The warning model [5] according to the driver’s cognitive status should be considered very important when designing advanced driver assistance systems (ADAS) and human machine interfaces. In our study, we made effective classifiers for warning model through machine and deep learning techniques to judge the outcome of the cognitive status of the elderly with electroencephalogram (EEG) signals. If this applies to human factors, such as EEG signal to the traffic sign recognition (TSR) of ADAS, it is expected to effectively contribute to the safe driving of the elderly.

## 2. Related Work

The design of TSR, one of the important subsystems of ADAS, has been a challenge for many years and has become an important and active research topic in the field of intelligent transportation systems [6]. Traffic Sign Recognition helps drivers to keep an eye on surrounding traffic conditions. In particular, quick understanding of traffic signs on highways is very important, and misunderstanding poses more safety concerns for elderly drivers. Allen et al. [7] tested for the same traffic signal recognition for all age groups, and a longer recognition and response time was required in the group of elderly subjects. David F. et al. [8] presented the results of a study that elderly drivers are more dangerous than young drivers at complex intersections of four directions due to their low peripheral perception. To improve this, they insisted that effective traffic signals and signs should be designed for elderly drivers without distracting vehicles having different directions.

Traffic signs include information that must be provided quickly to the driver before arriving in the affected area, so they must be intuitively designed to prevent driving distractions. In particular, the elderly with low cognitive ability can easily acquire information from a simple speed sign via a number, but complex signs with icons can lead to a lot of confusion. The driver’s focus on navigation to grasp traffic signals while driving can lead to a major accident. In order to overcome these gaze problems, HUD (head-up display), an advanced technology used in aircraft such as fighters in the past, is being spread to the market, starting mainly with luxury vehicles. Liu et al. [9] announced that HUD is helping to ensure safe driving by responding more quickly to changes in traffic signals by increasing perception of vehicle information regardless of speed. However, that study was not intended for the elderly. There is a possibility that they may experience difficulties while driving despite using the HUD due to a decrease in attention and concentration. Looking at major studies on the elderly, Halpern et al. [10] compared response times to sign and text traffic signals in two age groups (19–29 and 65–77 years old). Although the elderly responded relatively slower than the younger subjects, the response to the verbal traffic sign was 0.2 s faster on average than that of the symbol. There was no difference in the response time of the young subjects.

Elderly people have more difficulty in applying information technology (IT) than young people. In particular, IT engineers tend to underestimate the study of semiotics to improve cognition, so a lot of research in the relevant field is needed [11]. The Sha et al. [12] study showed that smartphone icons designed with monochromatic colors were more easily recognized by the elderly than icons designed with multiple colors. Few older people liked flat icons with more colors than solid flat icons for a colorful vision experience. As a result, the monochromatic verbal traffic sign is judged to be a more effective traffic sign for the elderly to focus on. However, research on various traffic signs for the elderly is very rare, and it can be very different in driving situations. Chae et al. [13] found a correlation between six basic emotions (happiness, sadness, anger, disgust, fear, and neutrality) and brain signals from head-up display (HUD) images. Brain activation information was obtained from the central sulcus, and concentration information was obtained from the temporal lobe. While 20 participants were exposed to 18 HUD images in a driving simulator, 16-channel EEG signals were obtained, and color was found to be a key factor in inducing emotion. Ito et al. [14] implemented a prototype of visual content for head-up display. As a result of an experiment using a driving simulator, it was confirmed that information sharing that includes visual content is effective for elderly drivers to trust and understand the system.

The event-related potential (ERP) refers to the potential difference that appears in response to an event such as a sensory or cognitive stimulus or movement in the brain, and is one of important technique in EEG research [15]. P300 which is the main ERP component occurs faster as attention and cognition is higher [16]. Liu et al. [9] showed that subjects respond more quickly when the contrast between the background color and the foreground color of traffic signs is high, and the content is simplified. As a result, a traffic sign with ’black foreground-green/blue background’ of a small contrast had greater P300 amplitude than the ‘black foreground-white/yellow background’ of big contrast. Major reactions were observed at P300 (300–400 ms) in centroparietal and parietal electrodes. Additionally, it is important to evaluate and quantify the individual P300 because it has differences for each individual depending on ages [17,18], the level of education [19], etc. There are many different methods to detect P300, from a simple method of detecting a positive peak through signal processing such as traditional ‘wavlet transform’ [20] to complex signal processing techniques such as PCA (principle component analysis) [21] and SVM (support vector machine) [22,23] having difficulty to implement. This point proves that P300 detection is an old important issue in EEG analysis, and there are many difficulties in detection.

The main purpose of our study is to analyze the types of traffic signs suitable for the elderly and to detect misunderstood brain waves to prevent accidents in elderly drivers. In particular, our study suggests using simple detection methods for ERP main components, and effectively made classification models with machine and deep learning of misunderstanding about traffic signs. We expect that the results of this study will contribute to preventing accidents, designing effective traffic signs, and improving the performance of the TSR system of ADAS for the elderly.

## 3. Methods

We expected young and elderly drivers to show different cognitive abilities. It was hypothesized that young drivers’ perception of traffic signals would be more sensitive. For this reason, both elderly and young adults were recruited as subjects. All participants were active drivers with a driver’s license, and 28 young and elderly were recruited. However, due to an error in data collection, the number of data in which all three experiments (triangle/circle/rectangle) were normally recorded during the collection consisted of 10 elderly and young people (see Table 1). The institutional review board (IRB number: 1040395-201610-01) at the Catholic University of Korea approved this study, and informed consent procedures adhered to institutional guidelines. Since most traffic signals are exposed to the driver for only a few seconds while driving, it is important to be aware of them very quickly so that the driver can respond quickly to driving behavior in various traffic situations. Representative traffic signals are classified as indication signs, regulatory signs, auxiliary signs, and road markings according to Korean laws. In this study, the changes in the elderly and young drivers’ EEG were measured when they differentiated main similar traffic signs of circles, triangles, and squares (see Figure 1).

In order to measure EEG, 64 channels of the NeuroScan instrument and Curry7 software were used. To minimize the noise of the recording, the impedance of the brainwave device was set to less than 5 kΩ before recording. The electrode attachment site used the internationally unified 10–20 method. This is a set of 10% and 20% intervals for the width and length of the head. Since EEG is sensitive to small movements, the subject was given an explanation of how to avoid it (e.g., movement, coughing, speaking, etc.). In order to minimize the factors that could affect the test subject, only the equipment necessary for the experiment were present, and the experiment conditions allowed the subject to concentrate as much as possible in an isolated dark area (see Figure 2).

Subjects were instructed not to drink too much and to get enough sleep before the day of the experiment. First of all, in order to reduce the noise caused by the subject’s movement as much as possible and to re-create the in-vehicle inside environment, a virtual HUD (head-up display) environment was created so that the subject could look ahead. One 24-inch monitor was placed in the front side, and another 14-inch monitor was placed at the bottom. The reflector glass of the bottom monitor was properly installed in consideration of the angle of reflection having been seen clearly like with HUD. Here, the 24-inch front monitor represents a road sign outside the vehicle and the reflector HUD represents a traffic signal displayed in the in-vehicle navigation. The total experiment was conducted for a total of 5 min in three sets of triangles/circles/squares for each subject, and each set was performed for 1 min and 40 s. Five pictures were shown randomly and repeatedly 20 times so that 75% of the pictures were different and 25% of the same pictures were displayed on two monitors. Whenever the traffic signal image changed, the Go/Nogo [24] method was used to click the left key of the mouse when a different picture comes out, and the right key when the same picture appears.

Each traffic signal used in the experiment was made to display for 0.2 ms, and a mouse click was used to determine the coincide between the traffic signal displayed on the HUD and the front side monitor within 1.8 s. After that, there was a delay of 3 s between the signals before another traffic sign came out (see Figure 3). Our study observed the comparison of the EEG signal of correct and incorrect answers’ as well as the comparison between the groups of the elderly and the youth. First of all, the range of correct answers indicated by left-clicking the mouse in cases when a different picture (75% of all questions) was displayed between HUD and the front monitor, and right-clicking when the same picture (25% of all questions) was displayed. In addition, the range of the incorrect answer excludes the range of correct answers and includes cases when the response was not answered within 1.8 s. A test program was developed with E-Prime (Psychology Software Tools, Inc., Sharpsburg, PA, USA ), which is famous for psychological experiments and research. In order to analyze the subject’s answer correctly, we made a unique number of questions and the subject’s left-/right-clicking of the mouse. In analyzing EEG, ERP components (n200/p300/n400/p600) were detected and analyzed through EEGLAB (v13.4.4b, Swartz Center for Computational Neuroscience, San Diego, CA, USA) and Matlab (R2014a, Mathworks, Natick, MA, USA). The machine learning model was made with Matlab, and deep learning was carried out using Tensorflow (v2.0.0, Google, San Francisco, CA, USA). According to the previous research [25], the central sulcus (emotion, movement, and intellectual abilities) behind the frontal lobe, the parietal lobe (spatial and sensory functions, when testing cognitive ability for shapes), and occipital lobe (visual function) are deeply correlated with recognizing shape. In consideration of these points, our study made a classification model with targeting mainly to analyze the central–parietal–occipital lobe as the main analysis target (see Figure 4).

In analyzing EEG signals, the study of ERP’s N200/P300/N400/P600 mainly focuses on two main factors: peak amplitude and latency. N200 is a negative-going peak wave in the vicinity of 150–350 ms, indicating the start of the event occurrence time. More generally, the N200 component has been characterized in identification of stimulus reflection [26], response selection timing [27], and detection of novelty or mismatch [28]. P300 is a positive-going peak wave [29] in the vicinity of 250–500 ms. The higher the attention, memory, and cognition, the wider the amplitude of the P300 and the faster the latency. In general, the peak amplitude of the P300 is related to stimulation information, and the peak latency of the P300 is not only related to the stimulation characteristics but also to the characteristics of the subject. In particular, when it perceives familiar pictures or letters, P300 is raised 300 ms from the point of stimulation, which shows that there is a very high correlation with the intimacy to stimulus experienced by each individual. In addition, P300 also occurs faster and the amplitude of P300 tends to increase when it has higher attention, memory, and cognition. It has the fastest P300 responsiveness in the twenties and becomes delayed with age. Likewise, the P300 event amplitude (μV) also reaches its maximum point starting in the 20 s, and tends to decrease with age [16].N400 is a negative-going peak wave generated around 250–600 ms, and is used in various studies including pictures [30], music [31], and syntax understanding [32]. N400 mainly reflects the process of searching long-term memory, and it is known that the amplitude of N400 increases as the recognition process deepens. P600 is a quantitative reaction observed in the vicinity of 500–800 ms. It is generated for grammatical errors or other linguistic anomalies [33] but is also used for interpretation in nonlinguistic fields such as music [34].

The original ERP signal (see Figure 5) has many fluctuations, much noise in the unnecessary band (e.g., 1-4 Hz DELTA wave or 30Hz or higher High BETA wave), many peaks (e.g., 3 to 4 times within 100 ms) within a short time due to the characteristics of the EEG signal. In this study, considering the characteristics of these signals, the following signal processing was selected and analyzed.

(1) Averaged original signal: The averaged signal of each person’s ERP signal.

(2) Smoothing: Smoothing signal with 20 ms time window (which causes less distortion chosen by experiments) with ‘(1)’.

(3) Zero-phase filtering: Band-pass filtering with signals of 4–30 Hz by zero-phase processing with ‘(1)’.

(4) Filtering and smoothing: Smoothing operation after signal processing with ‘(3)’.

Additionally, an automated method (see Figure 6) for detecting N200/P300/N400/P600 was proposed. First, the negative-going peak wave between 150 and 350 ms suggested by other studies was detected as the detection time point of N200. In general, most other studies mentioned that P300 is able to detect a positive-going peak wave in the range of 250 to 500 ms, but this range was not applicable in all cases. If N200 is detected in out of general range, the P300 also detects out of range. In preparation for this case, our method suggested finding the P300 point based on N200 with fixed gap time (FGT, 250 ms). If the appropriate maximum inflection point is not found in the reference range (250–500 ms) to find the P300, it is searched in the FGT range (see case 2 of Figure 6). This method that we propose is very effective in the following case studies 1–4 (see Figure 7, Figure 8, Figure 9 and Figure 10).

Even if the EEG signal has many fluctuations and abrupt phase change, our detection algorithm was clearly operated, and it showed a slightly different result according to the type of signal processing. In particular, all answer data regardless of generation need zero filtering and smoothing processing in order to detect exact ERP components such as N200/P300. In order to differentiate between correct and incorrect answers effectively, unsupervised machine learning was used first. In order to extract meaningful feature data previously, we separated for 1 s from N200 with upper (1)–(4) signal processing results. Afterwards, the elderly’s correct answer ERP data were clustered into 10 clustered groups with the representative expectation-maximization (EM) algorithm of unsupervised learning. The characteristic data (mean, standard deviation, and weight value) from EM were input into a Gaussian mixture model (GMM) to obtain the correct answer model. Each individual correct/incorrect data bit was input into the model, and the similarity was analyzed with the model and classified by log–likelihood.

In the second step, the multi-layer model (MLP) of supervised deep learning was used. In the same way, 1 s ERP data were cut from the N200. They were randomly trained with 75% among 30 correct answers and incorrect answers (three shapes of traffic signs for each of 10 young and old subjects), and 25% were used as test data with upper (1)–(4) signal processing results. As input data of correct/incorrect answers, 1 s ERP data were entered. The the total input layer of MLP has 70 nodes. After passing through the first hidden layer of 20 nodes and the second hidden layer of 10 nodes, a 1 node output layer was used. As a result, the learning model had a structure of ‘70-20-10-1’. The input/hidden layer used the rectified linear unit (ReLU) activation function, and the output layer used the sigmoid function. Since it is a binary classification problem that distinguishes correct/incorrect answers, the ‘binary_crossentropy’ is used as the error function and the ‘adam’ is used as the optimization function. The epoch value, which is the number of repetitions for the input value, was set to 500 times. The number of nodes in each layer and epoch value were set to have an optimal result without overfitting. In general, despite sufficient testing in deep learning, the test set is only 25% of the total data, so it is difficult to confirm whether it works correctly. This is especially true when the number of data is insufficient, such as in our study. We verified our classifier using k-fold cross validation [35] to overcome this point.

## 4. Result

### 4.1. The Result of Go/Nogo Test

The results of the subjects’ Go/Nogo test with E-Prime are as follows (see Table 2). As expected, the rate of incorrect answers of the elderly was 142% higher than that of the young. In particular, the elderly had a high error rate and deviation in the triangle sign. The circle and rectangle signs showed lower wrong answer rates than the triangle sign. To the contrary, youth had the lowest percentage of incorrect answers in the triangle signal. The circle and rectangle signs showed similar wrong answer rates.

### 4.2. Analysis of ERP Component Time

In the results of analyzing the event time of ERP components a confident correlation according to different traffic signs could not be found (see Table 3). However, as a whole, it was shown that incorrect answers have a delayed time compared to correct answers. In particular, the gap time between correct and incorrect answers was N200 (12 ms)/P300 (29 ms)/ N400 (14 ms) / P600 (−12 ms) for the elderly and N200 (32 ms)/P300 (31 ms)/N400 (−7 ms) / P600 (20 ms) for the young. In fact, our study expected that the elderly’s misunderstanding rate of traffic signs would be high, and these results demonstrate a correlation with the latency of the ERP main components. In particular, the N200/P300 showed the highest latency in elderly group. Additionally, traffic signs having a high error rate (e.g., triangle traffic sign of elderly) do not have any special correlation with ERP latency.

### 4.3. The Classification Results of Machine Learning

In order to distinguish between the correct answers and incorrect answers, the correct answer was modeled with GMM. Comparative analysis was performed through the log–likelihood of the correct answer and incorrect answer. The data of young adults was classified as correct/incorrect answers even without special signal processing, but the data of the elderly was indistinguishable. The elderly’s data were better differentiated as the signal processing process was added, but the youth’s data were not (see Table 4, Figure 11).

### 4.4. The Classification Results of Deep Learning

The correlation between incorrect answers and latency was confirmed with the main ERP data, and similarity was classified through GMM, but the data of the elderly were not clearly differentiated compared to the young, so a more effective classifier was needed. In particular, since the number of subjects was not large, effective classification and validation of deep learning were needed to accurately measure the results. Unlike the GMM model of young adults in which correct and incorrect answers were classified without any signal processing, the MLP classifier is bound to need the signal processing process for both the elderly and the young for better classification than GMM. Nevertheless, the MLP classifier has more efficient and accurate judgment results because it enables absolute comparison of correct–incorrect answers by the characteristic of supervised learning. Table 5 shows the results of evaluating the performance of the supervised deep learning (multi-layer model) classifier through k-fold cross validation. There were many variations according to the selection of training data by k-fold cross validation. In particular, the smoothing processing data showed the best performance as 75% accuracy for the elderly while the zero-phase filtering and smoothing processing data of the youth had 63% highest accuracy.

## 5. Conclusions

In this study, traffic signals misunderstood by the elderly were analyzed and classified using various methods: ERP, signal pre-processing, and artificial intelligence techniques. When a traffic signal was misunderstood, a delay mainly occurred in the P300 of all groups, but it was a subtle difference of 20–30 ms. Young people’s misunderstanding of traffic signals was able to be effectively distinguished through unsupervised machine learning, but not so for the elderly. They need an effective signal processing method such as zero-phase filtering and smoothing. On the other hand, supervised deep learning had a high accuracy of 75% with a simple smoothing processing using the data of the elderly subjects. In our study, the difference between the elderly and youth could be identified, and the misunderstanding of traffic signals by the elderly was efficiently detected. If more subject data is accumulated in the future, it can be effectively used in designing a customized ADAS system for the elderly.

## Figures and Tables

**Figure 1 sensors-21-04607-f001:**
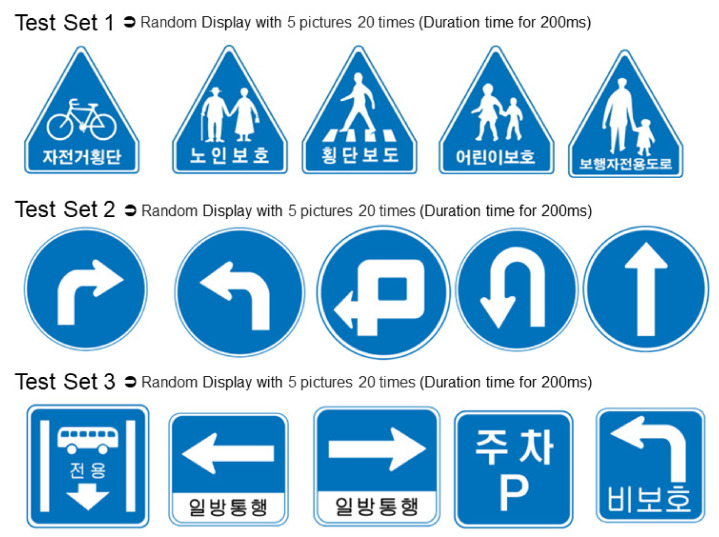
Major traffic signals: Three sets of quizzes were performed for each subject, and ERP results for each shape are presented in this study.

**Figure 2 sensors-21-04607-f002:**
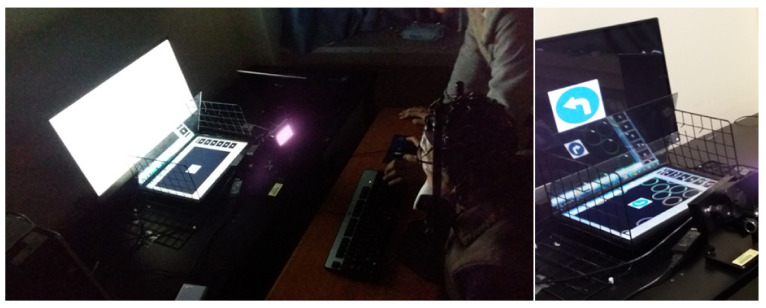
HUD display device constructed in the laboratory: Instead of the HUD device in the vehicle, a reflector was made with a simple fixing device with a glass plate, and the position of the face was adjusted and fixed for each individual so that two traffic signals could be clearly seen.

**Figure 3 sensors-21-04607-f003:**
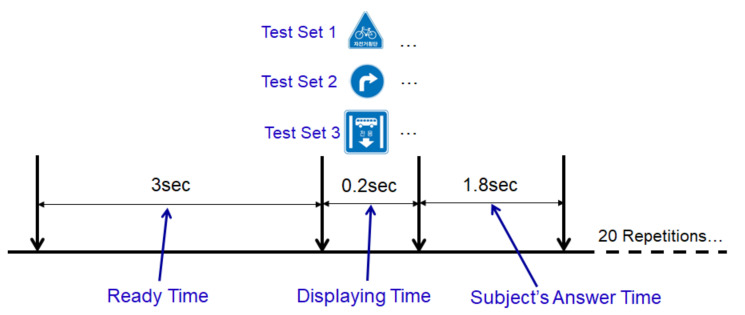
Subject traffic sign recognition sequence for each test item: Assuming a driving state, it was displayed on the screen for a short time of 0.2 s, and sufficient time was given to the next sign.

**Figure 4 sensors-21-04607-f004:**
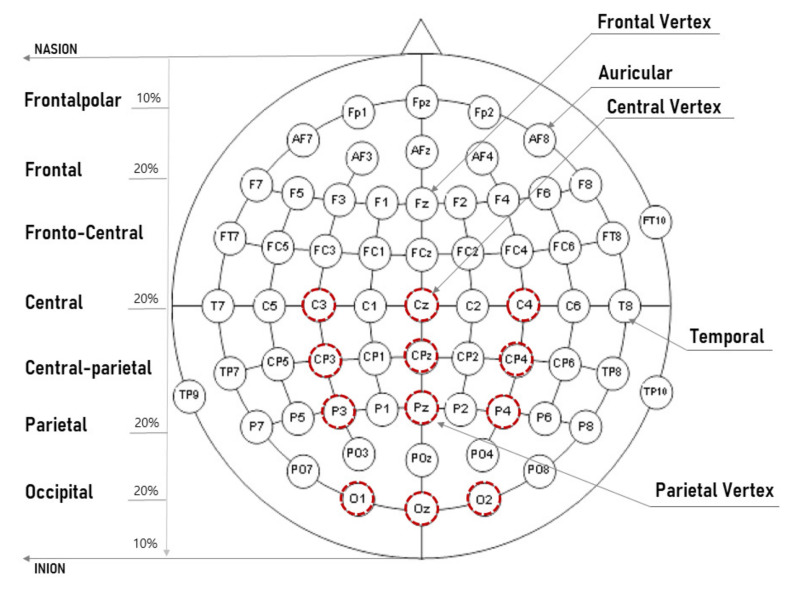
The red circle is the main analysis channel: The average value of the 12 channels was manipulated and analyzed.

**Figure 5 sensors-21-04607-f005:**
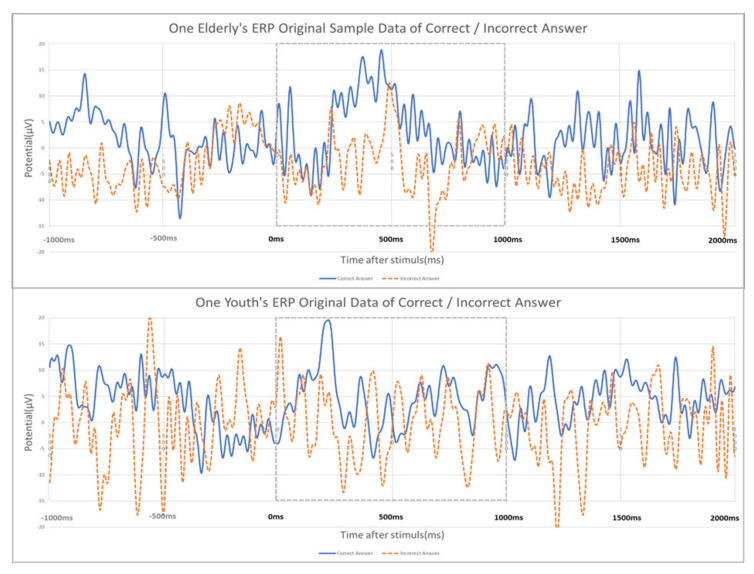
Original ERP (Cz, central vertex) data of an elderly/young subject (#1) from −1 to 2 s: In this study, the data from 0 to 1 s where the main components of ERP are located are the main targets.

**Figure 6 sensors-21-04607-f006:**
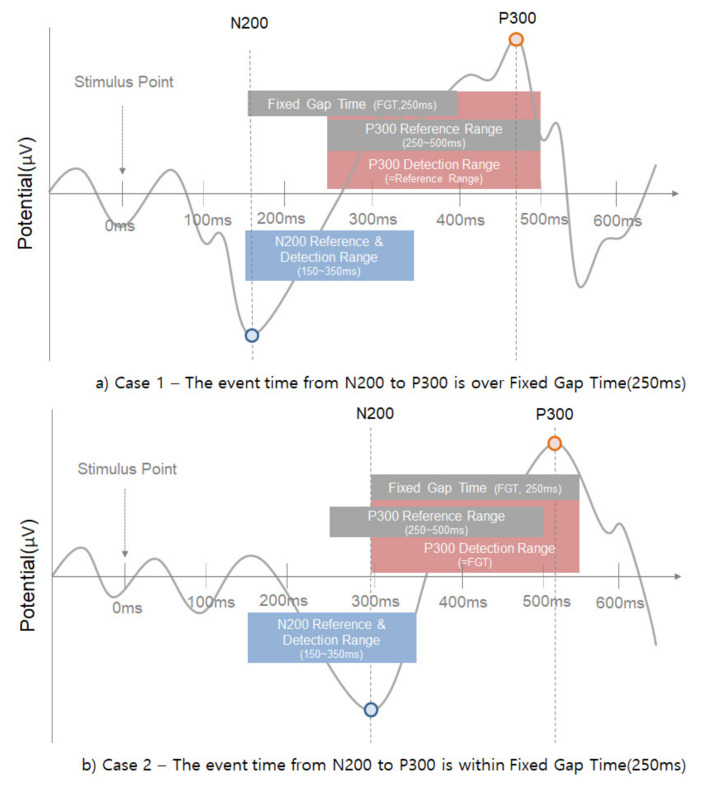
The proposed method in this study to find the P300 using fixed gap time based on the N200 point.

**Figure 7 sensors-21-04607-f007:**
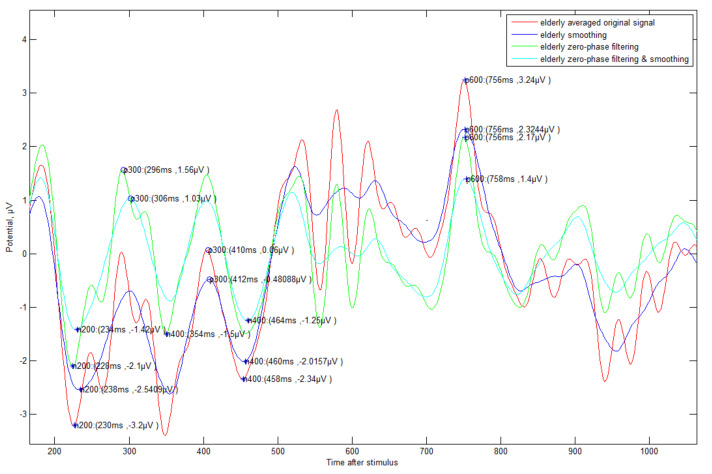
ERP Case Study 1—Elderly subjects (#9)’s correct answer about triangle symbol: There is a change in the time of detection of P300 ERP components according to various methods of signal processing.

**Figure 8 sensors-21-04607-f008:**
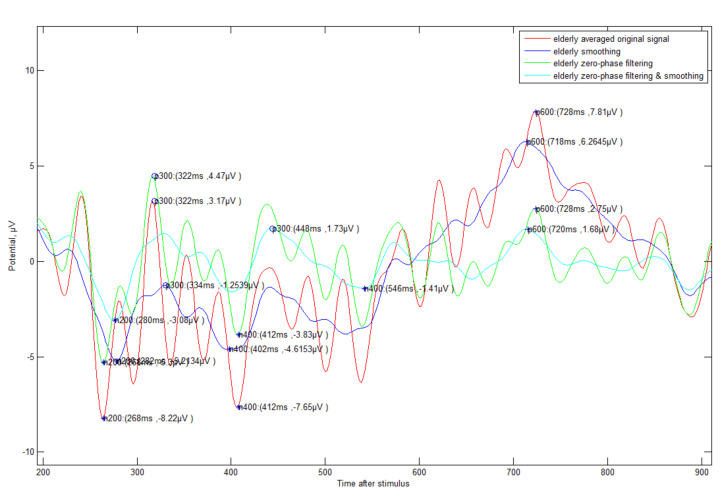
ERP Case Study 2—Elderly subject (#9)’s incorrect answer about triangle symbol: The fluctuation is more severe compared to the correct data, and the detection time is somewhat slower.

**Figure 9 sensors-21-04607-f009:**
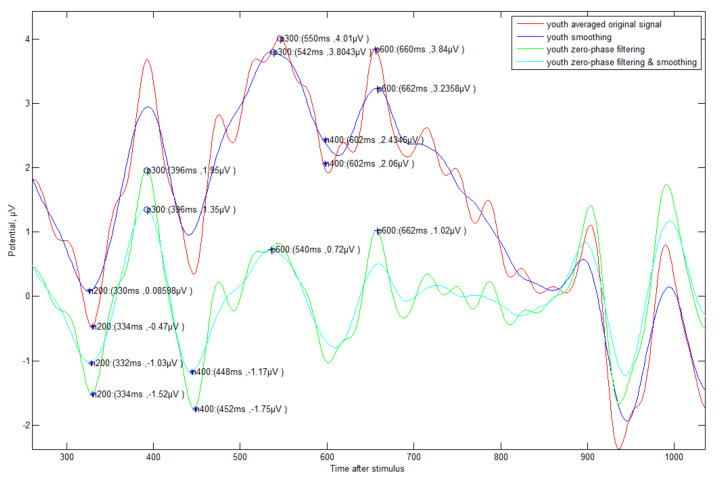
ERP Case Study 3—Young subject (#1)’s correct answer about triangle symbol: It can be seen that the viewpoint of each event component is faster than that of the elderly, and the fluctuation is small.

**Figure 10 sensors-21-04607-f010:**
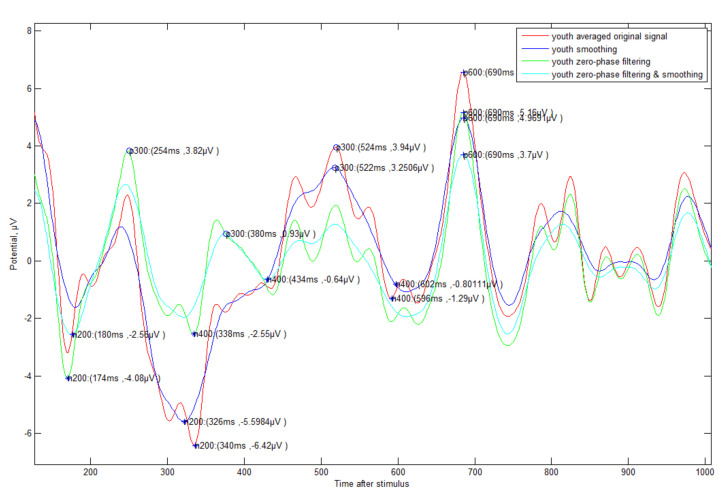
ERP Case Study 4—Young subject (#1)’s incorrect answer about triangle symbol: It is weaker than the elderly, but fluctuation is also more severe compared to the correct data.

**Figure 11 sensors-21-04607-f011:**
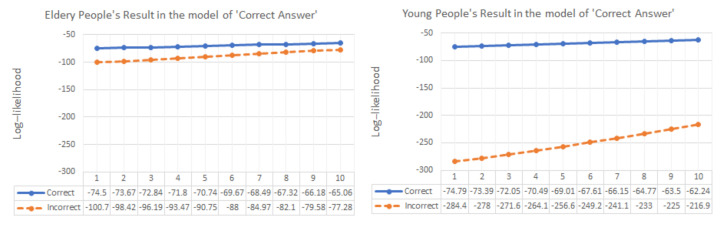
Log–likelihood result of original signal: It is difficult to distinguish the data of the elderly without signal processing.

**Table 1 sensors-21-04607-t001:** Age information of elderly and young subjects.

Variables	Elderly Group *n* = 10	Young Group *n* = 10	T Statistic	php-Value
	M (SD)	M (SD)		
Demographics Age (years)	66.2 (5.02)	22.7 (2.87)	22.59	0.0001

**Table 2 sensors-21-04607-t002:** Correct answer/incorrect answer rate for each traffic sign. Elderly subjects had a high percentage of incorrect answers, especially for the triangular sign.

Group	Shape of Traffic Sign	Correct Answer		Incorrect Answer		Error Rate
		AVG	SD	AVG	SD	
Elderly	Triangle	15.2	2.68	5	3.38	24.1%
	Circle	16.7	2.83	3.7	1.95	18.5%
	Rectangle	15.9	1.97	3.9	1.51	19.8%
	AVG	15.93	1.48	4.2	2.28	20.8%
Youth	Triangle	17.1	1.37	2.4	0.8	12.4%
	Circle	16.5	1.80	3.2	1.6	16.2%
	Rectangle	16.8	1.25	3.1	1.6	15.3%
	AVG	16.8	1.48	2.9	1.35	14.6%

**Table 3 sensors-21-04607-t003:** As a result of detection by zero-phase and band-pass filtering, there were differences by figure, but the difference between correct/incorrect answers was clearly seen in P300/P400 of the elderly and N200/P300 of the young.

Group	Shape of Traffic Sign	ERP Components Time(ms)							
		Correct Answer				Incorrect Answer			
		N200	P300	N400	P600	N200	P300	N400	P600
Elderly	Triangle	249	406	498	634	248	414	510	625
	Circle	241	376	490	668	232	420	517	618
	Rectangle	218	411	539	618	265	447	541	642
	AVG	236	398	509	640	248	427	523	628
	STD	13.13	15.48	21.21	20.82	13.64	14.61	13.20	10.07
Youth	Triangle	222	432	535	610	254	427	487	639
	Circle	203	356	480	595	236	403	479	617
	Rectangle	234	400	505	672	266	453	534	681
	AVG	220	396	507	626	252	427	500	646
	STD	12.55	31.08	22.65	33.27	12.50	20.44	24.56	26.80

**Table 4 sensors-21-04607-t004:** Comparison log–likelihood analysis result through GMM modeling of signal correct answer: The better the classification between correct answer and incorrect answer, the smaller the value.

Signal Processing Types	Subjects’ Log–Likelihood Difference Value (|Correct|–|Incorrect|)			
	Elderly		Youth	
	AVG	STD	AVG	STD
Averaged original signal	−19.1	4.6	−183.6	17.7
Smoothing	−19.6	0.7	−176.1	5.5
Zero-phase filtering	−345.1	142.2	−124.4	23.6
Zero-phase filtering and smoothing	−695.8	92.7	−75.1	1.4

**Table 5 sensors-21-04607-t005:** 4-fold cross validation results: The smoothing signal processing for the elderly and the zero-phase filtering and smoothing case for youth showed the highest accuracy.

Group	Signal Processing Types	Validation Accuracy					
		4-Fold Cross Validation				AVG	STD
		1	2	3	4		
Elderly	Averaged original signal	0.40	0.33	0.60	0.71	0.55	0.15
	Smoothing	0.53	0.80	0.87	0.64	0.75	0.13
	Zero-phase filtering	0.60	0.53	0.67	0.36	0.54	0.12
	Zero-phase filtering and smoothing	0.53	0.47	0.53	0.64	0.54	0.06
Youth	Averaged original signal	0.67	0.60	0.47	0.64	0.59	0.08
	Smoothing	0.53	0.47	0.47	0.50	0.49	0.03
	Zero-phase filtering	0.33	0.67	0.60	0.50	0.53	0.13
	Zero-phase filtering and smoothing	0.73	0.47	0.47	0.86	0.63	0.17

## Data Availability

Not applicable.

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
