# Peer review of "Traffic Sign Recognition Evaluation for Senior Adults Using EEG Signals"

_sensors, 2021, doi:10.3390/s21134607_

Round 1

Reviewer 1 Report

* Summary + Overall review
This paper presents a study to analyse the types of traffic signs suitable for the elderly and to detect misunderstood traffic signs through the analysis of brain waves. The authors compared the EEG signals from correct and incorrect answers and also between the elderly and youth groups. 

Although the paper is interesting and could have a real impact, I think it is still in a preliminary state, and needs to be further consolidated. As it is, I don't think the paper is ready to be accepted.

General comments:
- The study is well designed, but only 10 subjects in each group is too few.
- More details about the method used and the procedures used to achieve the results are required.
- The presentation of the different contributions of the paper is a bit confusing.

* Abstract
- The abstract can be improved to make it clearer what is the main goal of the paper. Is it the detection of misinterpretation of road signs, the study of ERP data from two populations, the creation of a machine learning model to detect incorrect answers, or both?

1. Introduction
- Again, the authors do not clarify what is the goal of the work. They provide a good context about the driving problems that the elderly face, but they do not identify the main objective of the work, summarise the work done, or present the main results achieved. (Only at the end of the related work the authors present the goal of their work)
- I suggest authors to revise the Introduction (and the abstract) to make it easier for the reader to understand what was done and with what goal.

3. Methods
- This section needs to be revised to provide more details about the several steps of the developed solution. In particular, more details about the signal analysis part, and about the unsupervised learning. It is not clear for me what was the information "stored" in the feature vectors used to describe each EEG signal, and then to perform clustering.

- The separation of the young and elderly data to create different models needs to be better explained.

- Is the method proposed for detecting N200/P300/N400/P600 a contribution of the paper? If yes, then the paper lacks related work about P300 identification.

4. Results
- Why the P600 is not reported in Table 3?

- In section 4.2, the analysis and discussion of the ERP components could be more detailed and complete.

- In section 4.4, authors only present the results achieved, without discussing them.

* Typos/Formatting issues
- Table 1: the headers for the elderly and young groups are swapped.
- pp6: "... clicking the left mouse when the different picture ..." vs pp5 "... left key of the mouse when the same signal comes out" - So, the left click was performed when the two images were equal or different?

Author Response

Please see the attachment
Sincerely Yours.

Reviewer 2 Report

The article shows the response of 10 young and 10 adult subjects using the EGG to different traffic signals. The following are the drawbacks associated with the study.

-In related work, the P300 terminology is used without having been previously presented. 
- The study focuses on the perception of signals using a HUD for application during the driving of a vehicle. However, during the experiment the only task performed by the subject is recognition. During driving the neural load is much higher than measured thereby having an effect on response times. To see how it affects driving they should have a workload similar to driving for example by simulating driving or at least watching a video of a vehicle on the road.
- The classification performed with deep learning is not explained in detail, no information on training, validation or sizing is given.
- The images used have a very low quality, its resolution should be increased.

Author Response

Please see the attachment.
Sincerely Yours.

Reviewer 3 Report

The present study is interesting and presents many elements of novelty in the field. 

Introduction section is well focused as the Discussion on results. 

Methods were described clearly.

Graphs and Figures are explicative really.

Please check and correct Table 1 because the demographics of the populations have been incorrectly reversed (eg Elderly people have the age of young group and vice versa). 

English is good. 

Author Response

(The authors gave the same response as above.)

Round 2

Reviewer 2 Report

I hope that future work will extend this work by taking into account the neural load associated with driving. Nevertheless, in my opinion, this work is suitable for acceptance.